# Position: Causal Machine Learning Requires Rigorous Synthetic Experiments for Broader Adoption

**Audrey Poinsot** [* 1 2]  **Panayiotis Panayiotou** [* 3]  **Alessandro Leite** [4 2]  **Nicolas Chesneau** [1]  **Özgür Şimşek** [3]
**Marc Schoenauer** [2]

## Abstract

Causal machine learning has the potential to revolutionize decision-making by combining the predictive power of machine learning algorithms with the theory of causal inference. However, these methods remain underutilized by the broader machine learning community, in part because current empirical evaluations do not permit assessment of their reliability and robustness, undermining their practical utility. Specifically, one of the principal criticisms made by the community is the extensive use of synthetic experiments. We argue, on the contrary, that **synthetic experiments are essential and necessary to precisely assess and understand the capabilities of causal machine learning methods**. To substantiate our position, we critically review the current evaluation practices, spotlight their shortcomings, and propose a set of principles for conducting rigorous empirical analyses with synthetic data. Adopting the proposed principles will enable comprehensive evaluations that build trust in causal machine learning methods, driving their broader adoption and impactful real-world use.

## 1. Introduction

Causal machine learning (Causal ML) uses Machine Learning (ML) algorithms to answer causal questions (Pearl & Mackenzie, 2018). Despite its transformative potential for decision-making, Causal ML has yet to achieve widespread adoption in the broader ML community. Indeed, because Causal ML methods are based on strong and often unfalsifi-

able assumptions, their practical value is often questioned. Practitioners argue that such assumptions are unrealistic for real-world applications, while ML researchers, accustomed to working with assumption-free methods, criticize the limited applicability of methods that require strict validation of such assumptions. In this context, Loftus (2024) argued that wide adoption of causal inference requires a mindset shift towards human-centered scientific pragmatism, prioritizing utility by striking the right balance between flexible hypothesis-free and "correct" (i.e., based on strong assumptions) models. However, a significant barrier to changing the community mindset is the limitations of current evaluation practices, which often fail to provide tangible evidence of the practical utility of causal methods and how they would perform under realistic conditions (Feuerriegel et al., 2024; Curth et al., 2024; Berrevoets et al., 2024).

In predictive ML, researchers have increasingly criticized current empirical practices, particularly the overreliance on predictive performance as the sole metric of success (Herrmann et al., 2024; Karl et al., 2024; Dehghani et al., 2021). They also argue that benchmarks often focus on narrow, well-defined tasks, failing to evaluate models' robustness, interpretability, and real-world applicability (Geirhos et al., 2020; Freiesleben & Grote, 2023; Longjohn et al., 2024). While these predictive-ML-oriented critiques offer valuable lessons for Causal ML, they need to be extended to meet the particular challenges of Causal ML empirical analysis.

A core challenge in evaluating causal methods stems from the fundamental problem of causal inference (Holland, 1986): for the same unit, it is impossible to observe the outcome under both conditions: when it has acted and when it has not. For example, in the case of vaccines, one can observe the outcome for individuals who have received the vaccine or for those who have not, but not both for the same individuals. This fundamental limitation means that counterfactual outcomes, as the basis of causal reasoning and describing any causal queries, remain unobservable. Consequently, evaluations of Causal ML methods rely predominantly on synthetic datasets (Geffner et al., 2024; Mahajan et al., 2024; Javaloy et al., 2023).

In addition to the possibility of accessing fully observable

---

[*]Equal contribution  [1]Ekimetrics, France  [2]TAU, LISN, INRIA Saclay, France  [3]Department of Computer Science, University of Bath, Bath, UK  [4]INSA Rouen Normandy, Normandy University, LITIS, Rouen, France. Correspondence to: Panayiotis Panayiotou <pp2024@bath.ac.uk>, Audrey Poinsot <audrey.poinsot@ekimetrics.com>.

*Proceedings of the 42$^{nd}$ International Conference on Machine Learning*, Vancouver, Canada. PMLR 267, 2025. Copyright 2025 by the author(s).

ground truth, synthetic data provides the advantage of controlled experiments, allowing researchers to rigorously evaluate methods within specific hypothesis settings. However, for many practitioners, the reliance on synthetic data constitutes a major adoption limitation, as such datasets often fail to represent the complexities of real-world scenarios (Gentzel et al., 2019; Curth et al., 2024; Berrevoets et al., 2024; Montagna et al., 2023).

We claim that synthetic data itself is not the root problem but rather how it is used in empirical analyses. Notably, **we argue that synthetic experiments are essential and necessary to precisely assess and understand the capabilities of causal machine learning methods.** Our work aligns with the positions of Loftus (2024) and Herrmann et al. (2024) for the general machine learning setting but extends them by addressing practical challenges associated with the analyses of Causal ML methods.

With this work, we aim to enrich the debate on rigorously evaluating Causal ML methods to improve current practices. We do believe that the most appropriate evaluation frameworks will emerge through collaboration of both the Causal ML community and the broader ML experimental design community, as both causal inference and evaluation expertise need to be combined. Thus, we see this paper as contributing to this interdisciplinary dialogue.

To support our position, we highlight key limitations in the current evaluation practices of Causal ML methods in Section 2, empirically demonstrate some of them through targeted experiments in Section 3, and propose a set of principles to carry out rigorous, reproducible, and rich empirical analysis using synthetic data in Section 4.

**Brief Background on Causal ML.** While predictive ML has transformed numerous fields by excelling at prediction and uncovering patterns in the data, many real-world challenges require causal reasoning (Pearl, 2009). Causal questions are classified in three cognitive levels by the Pearl Causal Hierarchy (PCH) (Pearl & Mackenzie, 2018). The first level reasons about associations and passively observed data. The second level investigates the effects of interventions (e.g., determining the effect of a policy change). Finally, the third level is about counterfactuals, reasoning about hypothetical situations that would have occurred under different interventions, as opposed to actual events. In contrast to predictive ML that answers first-level questions using observational data only, Causal ML answers causal questions, estimating a quantity of a higher level of the PCH using data from a lower one (Pearl & Mackenzie, 2018). However, lower-level data are almost never sufficient (Ibeling & Icard, 2020; Bareinboim et al., 2022). Hence, it is necessary to formulate explicit hypotheses about the higher level. This is why, in order to obtain stronger causal implications, Causal ML needs to take on more assumptions than predictive ML. This introduces a key theoretical property of causal questions: identifiability. A causal query is said to be identifiable if, under an appropriate set of assumptions, it can be answered from lower-level data—that is, the answer to the query exists and is unique. While identification provides a theoretical guarantee of the ability to express a causal query, it does not provide any information on the ability to estimate it in practice. Appendix A presents key concepts of causal inference and gives formal definitions of the technical terms used in this paper.

## 2. Limitations of Current Empirical Assessments of Causal ML Methods

**Problem 1: Ground Truth Data Are Scarce.** Collecting evaluation data for Causal ML is inherently more challenging than for predictive ML. In predictive tasks, ground truth labels can be directly observed, but ground truth causal queries often cannot be observed due to issues such as the fundamental problem of causal inference (Holland, 1986), confounding, and selection bias. Consequently, the community relies on expert knowledge (Chevalley et al., 2022; Sachs et al., 2005) or results of experimental studies (LaLonde, 1986; Shadish et al., 2008; Hill et al., 2004) for causal datasets.

Expert knowledge is expensive and scarce, requiring collaboration with domain experts to define valid causal models and assumptions. For example, Mooij et al. (2016) construct causal discovery datasets using common sense knowledge but their applicability is limited due to subjectivity and lack of formal guarantees.

Experimental studies, such as Randomized Controlled Trials (RCTs), are considered the gold standard to measure causal effects in real-world scenarios (Fisher, 1936; Cochran & Cox, 1948), providing a rigorous alternative. However, RCTs are expensive, time-consuming, and often ethically infeasible. In medical research, for instance, randomly assigning harmful treatments is not an option, forcing reliance on observational data. Even when conducted, RCTs face practical challenges such as non-random dropout (Greenland & Brumback, 2002; Tennant et al., 2021) and limited generalizability due to controlled settings, making them insufficient for training causal machine learning models, which require diverse data with broad covariate coverage. To address these difficulties, a few studies (LaLonde, 1986; Shadish et al., 2008; Hill et al., 2004) provide both observational and experimental datasets, enabling evaluation of causal inference methods across real-world and controlled conditions.

Also noteworthy is that no real dataset exists to evaluate Causal ML methods aiming at answering counterfactual questions, as counterfactual outcomes are inherently

unobservable (Holland, 1986). As a result, ready-to-use real datasets are missing. Moreover, relying on a few datasets coming from specific domains hinders the ability to draw general conclusions (Dehghani et al., 2021) or assess Causal ML capabilities across different applications.

**Problem 2: Synthetic and Semi-Synthetic Data Are (Unintentionally) Biased.** We refer to synthetic data as any data drawn from a fully artificial causal model such as a Structural Causal Model (SCM) or a Causal Graphical Model (CGM), defined formally in Appendix A. Synthetic data suffers from two sources of (unintentional) bias (Gentzel et al., 2019). The first source of bias arises from the fact that experiments are typically designed by researchers with specific goals and expectations—often to evaluate their own methods, hoping to demonstrate superior performance over existing approaches. Consequently, design choices may be influenced by these expectations, which can hinder comparability across studies (Poinsot et al., 2024; Cheng et al., 2022) and introduce biases in performance estimation, favoring certain methods (Curth et al., 2021). The second source of bias stems from the inherent limitations of synthetic data: it can incorporate only the features that researchers know how to model. As a result, "unknown unknowns" are inevitably excluded (Gentzel et al., 2019).

While these criticisms have largely been directed at synthetic data, semi-synthetic data are equally susceptible to the same biases. Semi-synthetic data are widely used in causal inference evaluation as an intermediary between purely synthetic and real-world data. They enable practitioners to evaluate Causal ML methods under controlled conditions while retaining certain characteristics of real-world data.

An example of semi-synthetic data is *causal discovery datasets*, which are constructed by fitting CGMs to real observational data while assuming a ground truth causal graph derived from expert knowledge (Lucas et al., 2005; Lauritzen & Spiegelhalter, 1988; Spiegelhalter et al., 1993; Beinlich et al., 1989; Onisk, 2003; den Bulcke et al., 2006; Schaffter et al., 2011; Smith et al., 2011; Göbler et al., 2024). These datasets inherit biases from both the choice of fitted CGMs and the assumptions embedded in the expert-defined causal graph. Similar concerns have been raised in other areas of ML, where dataset construction choices can implicitly favor certain methods, in particular, the ones using similar modeling assumptions (Acharki et al., 2023; Curth et al., 2024; Feuerriegel et al., 2024).

A similar challenge arises in *semi-synthetic datasets for evaluating CATE estimators*, where CGMs are again fitted to real data, under assumed structural constraints (Neal et al., 2020; Parikh et al., 2022; Athey et al., 2024; de Vassimon Manela et al., 2024). Here, the true Conditional Average Treatment Effect (CATE) is typically unknown and

is either derived from the learned CGM or specified by the user. This approach introduces a new issue in addition to implicit bias: the potential non-identifiability of the query of interest. If the dataset and assumptions do not satisfy the necessary conditions for identification, these methods will still converge to an estimate as if a unique solution existed (Petersen, 2024). This is particularly problematic in real applications, where identification assumptions are often unverifiable or even unfalsifiable (Ibeling & Icard, 2020; Bareinboim et al., 2022). Although this limitation is theoretically acknowledged (Parikh et al., 2022; Athey et al., 2024; de Vassimon Manela et al., 2024), no work has systematically evaluated the bias it introduces or how different methods might converge to specific representations. This is why we decided to experimentally illustrate, in Section 3.1, that such methods can induce unintentional bias in benchmarks. Another approach to constructing a semi-synthetic dataset for CATE estimation involves generating artificial observational datasets from an RCT (Keith et al., 2023; Gentzel et al., 2021; Hill, 2011; Zhang & Bareinboim, 2021). This is done by non-randomly sampling data points from the RCT to introduce confounding bias. In this case, the bias comes directly from the researchers' choices in the sampling strategy, which can significantly impact evaluation outcomes.

Finally, *synthetic outcome datasets* (Curth et al., 2021; Dorie et al., 2019; Shimoni et al., 2018; Hill, 2011) modify real observational datasets by replacing the outcome variable with a synthetic value generated from a fully synthetic causal mechanism. Although this approach ensures that the ground truth CATE is known, it introduces the same biases as fully synthetic data, as the generated outcomes reflect researcher-imposed assumptions rather than true causal mechanisms.

In summary, researcher design choices (whether in defining synthetic mechanisms, selecting sampling strategies, or modeling causal effects) introduce biases that affect both synthetic and semi-synthetic datasets. These biases limit comparability across studies and can distort performance evaluations of Causal ML methods.

**Problem 3: Synthetic Experiments Lack Sufficient Complexity to Encourage Adoption of Causal ML.** Synthetic experiments used to evaluate Causal ML methods are frequently criticized for their over-simplicity (Curth et al., 2024; Poinsot et al., 2024; Cheng et al., 2022; Gentzel et al., 2019). First, synthetic experiments are typically derived from overly simplistic causal models. For instance, additive noise models (Hoyer et al., 2008), despite concerns about their suitability for empirical evaluation (Reisach et al., 2021), remain widely used (Mahajan et al., 2024; Bach et al., 2024; Huang et al., 2024; Curth & Van Der Schaar, 2023; Javaloy et al., 2023; Acharki et al., 2023). Addition-

ally, several recent works further restrict their analyses to quadratic (Huang et al., 2024; Curth & Van Der Schaar, 2023) or generalized linear causal mechanisms (Mahajan et al., 2024; Bach et al., 2024).

Secondly, synthetic experiments often lack sources of randomness, i.e., some simulation parameters, such as the causal graph or level of confounding, are fixed, which restricts the scope of the analysis. Although some efforts have been made to introduce randomly generated causal models for evaluation (Rudolph et al., 2023; Gupta et al., 2023; Kalainathan et al., 2020), this practice has not yet been widely adopted (Poinsot et al., 2024). Finally, robustness analyses are largely neglected. Causal ML methods are typically evaluated on datasets that strictly adhere to all the assumptions required by the method itself (Gentzel et al., 2019; Hutchinson et al., 2022; Petersen, 2024; Bouvier et al., 2024), offering little insight into real-world applicability under imperfect conditions. In Section 3.2, we use experiments to illustrate how robustness analysis can provide valuable information to practitioners and researchers.

We acknowledge the value of simple, didactic examples in research papers. They help clarify theoretical contributions. However, such examples alone cannot serve as evidence of a method's general utility. This lack of rigorous evaluation contributes to reluctance to adopt Causal ML methods by practitioners, who may consider these methods to perform well only in idealized settings.

**Summary and Impact on Adoption.** Real-world datasets for evaluating Causal ML methods are scarce due to the high cost, technical complexity, and ethical concerns of collecting experimental data. As a result, the community primarily relies on synthetic and semi-synthetic experiments to assess model performance. However, these evaluations suffer from two key limitations. First, the high degree of freedom inherent in (semi-) synthetic experiments introduces biases that can distort conclusions and hinder comparability across studies. Secondly, the synthetic evaluations often lack sufficient realism to encourage the adoption of Causal ML. These two shortcomings are direct consequences of how synthetic experiments are currently designed and analyzed. We do not claim that rigorous synthetic evaluation alone will guarantee the broader adoption of Causal ML. Rather, we argue that such evaluation is a necessary step for its adoption by ML researchers and practitioners.

## 3. Empirical Demonstrations of Problems in Causal ML Evaluation

The experiments[1] in this section demonstrate how the problems identified in Section 2 can arise in practice. We do not provide an explicit experiment for Problem 1 (Section 2)

related to real-data scarcity since that constraint, by definition, cannot be bypassed experimentally. Instead, we focus on bias in semi-synthetic experiments as discussed in Problem 2 (Section 2) and the lack of complexity in synthetic experiments as discussed in Problem 3 (Section 2). Our experiments spotlight key issues rather than provide exhaustive evaluations or benchmarks.

### 3.1. Demonstrating Problem 2: Examining Bias in Semi-Synthetic Datasets

Curth et al. (2021) demonstrated that experiments using synthetic outcomes can be biased. To do so, they altered the IHDP semi-synthetic benchmark (Hill, 2011) by modifying the outcome functions to make the treatment effect linear—the relationship between treated and untreated outcomes was changed while leaving the covariates identical. This small change in the data-generating process significantly shifted the relative method rankings. Their findings underscore how seemingly minor design choices in synthetic and semi-synthetic datasets can reverse conclusions about method performance (including which method is "best"), highlighting implicit biases from the choice of data generative process.

Following the same logic as Curth et al. (2021), we demonstrate that semi-synthetic methods, such as RealCause (Neal et al., 2020), which fit causal models to real data, can also introduce systematic biases. While other works, such as those by de Vassimon Manela et al. (2024), Athey et al. (2024), and Parikh et al. (2022), also investigate generative evaluation techniques, we focus on RealCause because it has been widely used in recent research for evaluating causal inference methods (Pros & Vitria, 2024; Bozorgi, 2021; Ter-Minassian et al., 2024; Shoush & Dumas, 2024; Mahajan et al., 2024; Zhang et al., 2024; van der Laan et al., 2024), making it an ideal candidate for illustrating the potential limitations of current generative evaluation methods. RealCause creates new "realistic" datasets with user-specified causal effects by training a generative model on existing input data. These generated datasets are intended for benchmarking causal inference methods. In this experiment, we investigate whether such benchmarks may implicitly favor certain approaches over others.

First, we replicate the Average Treatment Effect (ATE) error on the IHDP dataset reported in the original RealCause paper and confirm a point-estimate ATE error of $0.17$, compared to the true ATE of $4.02$. However, when we fix the realization used in the original experiment and vary the random seed (20 seeds for a single realization), we observe a mean ATE error of $0.38 \pm 0.39$, with some seeds yielding errors as high as $1.77$. In contrast, when we fix the seed

---

[1]The code for our experiments can be found at https://github.com/panispani/causalml-needs-synth-eval

(e.g., seed 123) and vary the realization (across 100 realizations), the error increases substantially, with a mean of $0.95 \pm 1.36$, and some realizations yielding errors as high as 9.45, leading to estimated ATEs that are multiple times larger than the true value. Finally, when computing the mean ATE error for each of 100 realizations (each averaged over 20 random seeds), we find extreme variability; in one case, the error reaches $6.209 \pm 11.318$, with the true ATE being $-0.604$. More details are provided in Appendix B.

These results reveal two fundamental issues: high error and extreme variance, both across seeds and across dataset realizations. Relying on a single seed conceals this variability, making benchmarks fragile and potentially misleading. More critically, the ranking of causal inference methods produced by RealCause is itself unstable—what appears to be the best method under one seed or one realization may rank among the worst under another. This instability, combined with a high ATE error for some realizations, raises a serious concern: RealCause does not merely introduce noise; it could systematically bias rankings favoring methods that align with the errors induced by its own generative assumptions. Hence, if one wants to use RealCause responsibly, further experiments need to be performed to understand which features of the datasets make RealCause more or less stable and accurate.

### 3.2. Demonstrating Problem 3: Testing Beyond the Identification Domain

As noted in Problem 3 (Section 2), synthetic evaluations often use data generation processes that align closely with the assumptions of the tested methods, potentially obscuring their weaknesses when these assumptions do not hold. In this section, we use Causal Normalizing Flows (CausalNF) (Javaloy et al., 2023) to show how studying methods beyond their identification domain can yield valuable insights for practitioners and researchers. We selected CausalNF because it is a state-of-the-art Causal ML method for counterfactual estimation. As a result, its evaluation relies exclusively on (semi-) synthetic experiments. While many other approaches could be explored, we focus on CausalNF to illustrate our perspective, leaving broader evaluations for future work. Full experimental details can be found in Appendix C. CausalNF has been developed under a strong assumption: the causal mechanisms of the SCMs generating the data are diffeomorphic (i.e., differentiable and bijective with differentiable inverses). By deliberately violating these idealized conditions, we reveal potential limitations and robustness of CausalNF, opening up new questions and highlighting the importance of realistic evaluations.

**When Assumption Violations Do Not Affect Performance.** For the first experiment, we adopt the

Triangle$_{NLIN}$ SCM from the original paper (Javaloy et al., 2023) and slightly change the original causal mechanisms. We apply a transformation to the noise—either a segmented linear function or a sinusoidal function—deliberately violating the diffeomorphism assumption while preserving all other aspects of the structural equations. The results (see Appendix C) show that CausalNF remains robust against these specific violations: root-mean-square errors for counterfactual predictions did not significantly increase. This result should not be interpreted as concluding that "diffeomorphism violations never matter." Instead, it demonstrates how method-friendly choices in synthetic datasets (e.g., minimal noise-parent interactions) can allow a method to perform well despite violations.

**When Assumption Violations Deteriorate Performance.** For the second experiment, we test CausalNF on a non-identifiable counterfactual example by Nasr-Esfahany & Kiciman (2023), where CausalNF is trained on two distinct SCMs sharing the same observational distribution but different counterfactual ones (see Appendix C). The results show that CausalNF consistently defaults to learning one variant of the two structures, resulting in large errors whenever the true variant is the alternative. Although it is theoretically expected that the algorithm cannot distinguish these two variants with observational data alone, we discover here that CausalNF tends to converge to only one of them, leading to a consistently biased error.

**Conclusions.** While this study is not exhaustive, it underscores the importance of conducting thorough evaluations beyond the standard identification domain of Causal ML methods. As demonstrated with CausalNF, behavior under assumption violations is difficult to predict. Such evaluations serve two key purposes. First, when empirical evaluations show robustness properties, they can guide theoretical advancements by consistently identifying properties or structures that perform well under specific conditions. Secondly, when they show failure modes, they enable practitioners to better understand the circumstances under which models can be trusted and where they might fail, supporting informed decision-making and encouraging broader adoption of Causal ML methods.

## 4. Principles for Rigorous Causal ML Synthetic Evaluation

This section develops a set of concrete and actionable principles for evaluating Causal ML methods through synthetic experiments. These principles are designed to empirically assess the utility of Causal ML methods, specifically their ability to answer causal questions from a dataset under a particular set of assumptions. To achieve this, a Causal ML method must be rigorously evaluated (cf. Principle 1) using different indicators (cf. Principle 3), in a methodical man-

ner (cf. Principle 4), regarding its ability to answer causal questions under different sets of assumptions (cf. Principle 2).

### 4.1. Principle 1: Synthetic Data Is Necessary to Derive Rigorous and Precise Conclusions

First, as highlighted in Problem 1 (Section 2), synthetic data is the only reliable source of ground truth for causal queries given that real-world datasets cannot provide access to counterfactual outcomes. Without ground truth, it is hard to objectively measure the accuracy of Causal ML methods. Secondly, synthetic experiments allow full control over the data-generating process, enabling researchers to systematically vary parameters such as noise levels, confounding, or structural complexity. Hence, they enable researchers to run randomized controlled experiments for more comprehensive experiments, rigorously assessing how specific factors influence a method's performance—this is unattainable with real or semi-synthetic datasets. Finally, relying on real or semi-synthetic data ties evaluations to the idiosyncrasies of specific problems, limiting the scope and reliability of conclusions.

While we argue that synthetic data is necessary for rigorous evaluation, we do not claim that synthetic data alone is sufficient or the only valid way of evaluating a Causal ML method. For instance, semi-synthetic data can introduce a degree of realism that is valuable in certain contexts. However, such data also originates from Causal ML generation methods, which must themselves be rigorously evaluated before they can be relied upon. Our concern centers on temporality rather than significance: we argue that semi-synthetic data generation methods should first be rigorously evaluated using synthetic data to identify their limitations. Only after this evaluation can these methods be more widely adopted, ensuring that their advantages are leveraged while being aware of the potential biases inherent in them.

### 4.2. Principle 2: Synthetic Design Choices Must Be Clearly Stated to Mitigate Unconscious Bias

As shown in Problem 2 (Section 2), one major limitation of (semi-) synthetic experiments relies on the fact that the researchers' high degree of freedom in experimental design choices leads to biased results and conclusions. We argue that bias is not necessarily unwelcome given that finding the overall most performing method is, in general, not relevant to researchers or practitioners (Lim et al., 2000; Belkin et al., 2019; Liao et al., 2021; Hutchinson et al., 2022; Lones, 2024). Instead, we argue that unconscious, unknown, or untracked bias is problematic because it increases the risk of drawing incorrect or misleading conclusions. One way to mitigate this issue is to make all experiment design choices transparent, to be clear on the domain of validity of the

drawn results. The need for transparency of experimental conditions for reproducibility and rigor has long been recognized in the experimental sciences as well as in the ML community (Gigerenzer, 2018; Calin-Jageman & Cumming, 2019; Marie et al., 2021). In this section, our goal is to clarify what it concretely means for synthetic experiments on Causal ML methods. To make our point more specific, we propose the following classification of design choices.

We claim that any Causal ML method experimentation should at least make explicit the following five elements: (i) the **set of causal models** studied, (ii) the **set of causal queries** of interest, (iii) the **set of training data**, (iv) the **generation algorithm** producing the synthetic causal models, queries, and datasets, and (v) the **distribution** that the generation algorithm implies on the space of synthetic examples defined by the elements (i), (ii), and (iii). These five components must be described in such a way that anyone can reproduce the same synthetic experiments and interpret the results without having to make any additional assumptions. An example of design choice description following this classification is given through the CausalNF example later in this section.

In particular, the set of causal models must be defined by a conditional expression. The set of queries must specify the causal quantity studied, its PCH level, the variables, and the values considered. The set of training data must be defined not only by the dimension of the dataset but also the level of the PCH to which it belongs, as well as any potential perturbations it may have undergone (such as measurement error, selection bias, hidden variables, presence of outliers).

All too often omitted, the algorithm for generating synthetic examples and the distribution it induces on the space of synthetic examples are crucial elements to describe. Indeed, there exist many ways of generating synthetic examples. One may decide to sample synthetic examples according to a predefined distribution. This approach requires using a perfect sampling algorithm to guarantee that the distribution of the sampled elements corresponds exactly to the desired upstream distribution (Propp & Wilson, 1998; Fotakis et al., 2022). Although progress has been made in this field, notably on perfect sampling of Directed Acyclic Graphs (DAGs)s (Talvitie et al., 2020; Harviainen & Koivisto, 2024), as far as we know, no perfect algorithm exists for sampling elements as complicated as a set of causal models, causal queries, and datasets. This is why one may also decide to explore the synthetic space rather than sample elements from it. Using a random walk, a grid search, or even optimizing for a predefined criterion are possible options. However, each exploration strategy may induce different distributions over the space of synthetic examples, which can have a major influence on the results. Indeed, when aggregating results (e.g., mean, median), if the dis-

tribution of the synthetic elements considered is different, the aggregated quantities and the conclusions derived from them could be very different. Hence, not accounting for the effect of the distribution creates a potential bias that remains unaccounted for in the analysis of the experiments. This is why, to derive rigorous analyses, the distribution considered over the space of synthetic examples needs to be clearly presented. In particular, if no expression can be derived from the exploration algorithm to describe the distribution, the latter should at least be evaluated empirically.

**CausalNF Example.** Assume that one wants to extend the experiments carried out in Section 3.2 on the robustness of CausalNF (Javaloy et al., 2023) against violation of the bijectivity assumption. One could, for example, define the following synthetic experimental design choices (see Appendix D for a formal description):

**(i) Set of causal models:** The SCMs such that their causal graphs are DAGs, their causal mechanisms are fully connected neural networks, and their distributions of the exogenous variables are uniform over $[0, 1]$.

**(ii) Set of queries:** The counterfactual distributions of any endogenous variable, under the intervention of any other variable, to an observed value of the training set, given the factual realization of any sample in the training set.

**(iii) Set of training data:** The observational datasets of size 1000 are drawn from the entailed distribution of the set of SCMs without perturbation.

**(iv) Generation algorithm:** (1) Uniform sampling of a set of control parameters (i.e., the parameters defining the SCMs, queries, and datasets), (2) uniform sampling of a DAG (Talvitie et al., 2020) given the set of control parameters, (3) initialization of the neural networks weights and bias following the Glorot uniform (Glorot & Bengio, 2010) given the set of control parameters resulting in a fully defined SCM, (4) dataset creation from sampling 1000 observations from the SCM, (5) query creation by randomly picking the outcome and intervened variables from the endogenous variables, and (6) randomly sampling from the training set the intervention value and factual realizations.

**(v) Induced distribution:** The joint distribution of the control parameters of the SCMs, queries, and datasets is, by definition, multivariate uniform with independent components. However, the distribution of observable characteristics, such as the bijectivity of the causal mechanisms or the existence of interaction between endogenous and exogenous variables, is not straightforward to determine. Hence, for the characteristics that may be relevant to consider when studying the behavior of CausalNF when the bijectivity assumption is violated, it will be necessary to carry out an empirical analysis of their distribution by analyzing the SCMs sampled by the generation algorithm.

### 4.3. Principle 3: Going Beyond Aggregated Accuracy in the Identification Domain with Comprehensive Experiments

**Evaluating Across and Beyond the Identification Domain.** Herrmann et al. (2024) and Karl et al. (2024) emphasize the importance of designing empirical research beyond narrowly defined contexts to probe robustness and generalization across diverse settings. This is particularly relevant to Causal ML, where models often rely on strong assumptions within their identification domains. To fully evaluate their capabilities, methods must also be tested in scenarios that challenge these assumptions, such as shifts in causal structures. Evaluating models both within and beyond their identification domains exposes failure modes and provides insights into their limits under real-world conditions. Therefore, we encourage researchers to define large synthetic experimentation spaces, i.e., large sets of causal models and datasets as defined in Principle 1 (Section 4).

**It Is Not Only About Accuracy.** An overemphasis on predictive performance, as criticized by Karl et al. (2024), leads to unhealthy research incentives that prioritize superficial improvements over meaningful progress. Notably, reducing evaluation to accuracy risks ignoring critical dimensions such as robustness, scalability, stability, and interpretability (Freiesleben & Grote, 2023; Geirhos et al., 2020; Crabbé et al., 2022). This is all the more important for Causal ML given its growing use in sensitive decision-making domains. Moreover, practical considerations such as computational efficiency, cost of data collection, and the cost of validating causal assumptions must be integrated into evaluation frameworks. A shift toward more comprehensive analyses will ensure that Causal ML methods are not only theoretically sound but also practically viable in diverse and uncertain real-world scenarios.

**Focusing on Insights Rather Than Aggregate Performance Metrics.** Herrmann et al. (2024) advocate for "insight-oriented exploratory research", which prioritizes understanding over mere incremental performance improvements. Focusing solely on performance metrics can lead to biased evaluations and a misleading perception of progress (Balduzzi et al., 2018). This perspective is essential for Causal ML, where the scarcity of diverse, representative datasets makes it problematic to rely solely on aggregate performance metrics for model selection. By systematically documenting negative results, exploring edge cases, and analyzing failure scenarios, researchers can uncover deeper knowledge that drives theoretical and practical advancements (Karl et al., 2024). Such practices promote a culture of transparency and scientific rigor, shifting the focus from benchmarking competitions to discovering meaningful insights.

**Capturing Real-World Complexities with Synthetic Experiments.** To manage the complexity of simulations that aim to mirror real-world conditions, we recommend a progressive approach to increasing simulation complexity. Begin with a simple, intuitive experiment, e.g., a small graph, simple mechanisms, binary variables, sufficient data, and no assumption violations. Gradually introduce one new complexity at a time, e.g., hidden confounders, selection bias, small datasets, or mixtures of SCMs. For each added complexity, carefully analyze its impact on the method's performance before adding the next one. This approach allows for the isolation of the effects of previously added complexities while progressively increasing the overall experiment sophistication. Additional complexities to consider include the presence of outliers, violation of the ignorability assumption, erroneous causal graph, or adjustment sets.

### 4.4. Principle 4: Developing Standardized Evaluation Frameworks to Promote Best Practices

As discussed in Problem 2 (Section 2), (semi-) synthetic evaluations of Causal ML methods have limited comparability. Such a lack of consistency undermines both replicability and the ability to draw meaningful conclusions across studies (Herrmann et al., 2024). Standardized evaluation frameworks are critical in addressing this challenge. Indeed, by providing a structured approach to making design choices for researchers, standardized evaluation frameworks favor thorough documentation of data generation processes, assumptions, and potential biases. They also enable researchers to extract, share, and reproduce design choices easily (Dehghani et al., 2021).

There exist two open-source platforms for benchmarking Causal ML methods: CauseMe (Runge et al., 2019) and CausalBench (Kapkiç et al., 2024). CauseME is dedicated to causal discovery in time series data; CausalBench encompasses a broader scope of causal tasks, such as treatment effect estimation. By providing pre-coded datasets, models, metrics, and evaluation algorithms, they make evaluation procedures more transparent, fair, and easy to use. Although these platforms represent fundamental and valuable work to promote the standardized evaluation of Causal ML methods, much work remains to be done to implement all the recommendations of the three principles introduced in Sections 4.1 to 4.3. For instance, neither platform currently includes counterfactual estimation tasks. In addition, information on datasets, particularly (semi-) synthetic ones, lacks sufficient details to comply with Principle 2 (Section 4).

This is why we encourage the community to continue this effort and enrich the existing frameworks. However, we would like to highlight two critical concerns. First, it is important to avoid overly narrow standardization, where benchmarks unintentionally encourage models to exploit dataset artifacts, undermining generalization (Geirhos et al., 2020; Curth et al., 2021). Secondly, it is essential not to multiply the number of frameworks and platforms, which would run counter to the goal of homogenization. We therefore encourage extending and refining existing frameworks wherever possible rather than developing new specialized frameworks. Ultimately, in order to assess Causal ML methods, evaluation frameworks need to balance consistency and flexibility, promoting open collaboration while accommodating diverse metrics and perspectives. By doing so, they can facilitate the adoption of best practices and accelerate meaningful progress in Causal ML research and its adoption by the broader ML community.

## 5. Limitations of Our Position

While the principles outlined in Section 4 address significant gaps in current evaluation practices of Causal ML methods, several challenges remain that require careful consideration and ongoing effort to overcome.

One key challenge lies in achieving **adherence to the proposed principles** by the Causal ML community. The success of any standardized approach hinges on collective effort and widespread application within a community. Without such collaboration, these principles risk remaining underutilized, limiting their potential to drive comprehensive evaluations, which are necessary for a broader adoption of Causal ML by the machine learning community. Encouraging compliance with the principles will require not only consensus-building but also clear demonstrations of the developed framework's value across diverse applications.

Another consideration is the **resource intensity** to implement the principles. Rigorous evaluation requires significant computational resources, which may be unavailable for researchers and organizations with constrained budgets. Addressing this limitation calls for innovation in resource-efficient methodologies, ensuring that the benefits of these practices are broadly available without creating barriers to participation.

Finally, synthetic data, while indispensable for controlled experimentation, has an inherent limitation: its **inability to capture unknown unknowns**—phenomena or dynamics outside the scope of the data-generating process (Gentzel et al., 2019). This limitation underscores the importance of complementing synthetic data evaluations with real-world experiments and alternative methodologies to ensure comprehensive and robust assessments ( as discussed in Section 6).

Acknowledging these challenges does not diminish the importance of rigorous synthetic evaluation but highlights the collective work required to fully realize the potential of Causal ML and ensure its broad adoption.

## 6. Alternative Views

Throughout this paper, we showed why synthetic experiments are necessary and how to use them appropriately to rigorously evaluate Causal ML methods. However, because of their intrinsic inability to model unknown unknowns (Section 5), synthetic experiments are insufficient to completely assess real-world performance. While semi-synthetic experiments are convenient for building synthetic data similar to real ones, they also suffer from the same modeling limitation, see Problem 2 (Section 2). As a result, one could argue that **the community should instead focus on real experiments because of their realism and unique ability to encapsulate unknown unknowns**. Although this alternative vision is orthogonal to our position, it is also complementary and, therefore, of great interest.

The major difficulty with real experiments resides in the fact that the amount and diversity of real datasets to evaluate Causal ML methods are limited, as discussed in Problem 1 (Section 2). Indeed, controlled experiments (Fisher, 1936; Cochran & Cox, 1948) are resource-intensive, sometimes unethical, or even impossible to implement in practice. Below, we propose a number of research directions aiming to reduce this limitation.

First, the community could put more effort into gathering more RCTs data coupled with observational datasets (Curth et al., 2024; Cheng et al., 2022). An important consideration would be to prioritize the diversity of existing benchmarks. Today, biology and social sciences are overrepresented, while physics, for instance, is underrepresented. In addition, designing an experiment requires knowledge of the domain of interest to make sure the unconfoundedness assumption holds. There is a need for interdisciplinary collaboration (e.g., across healthcare, engineering, and economics) to collect diverse, high-quality datasets that reflect real-world complexities.

Secondly, finding new ways to exploit information sources that are less rigorous than RCTs, such as quasi-experiments (Angrist & Pischke, 2009) or unproven expert knowledge, is also a promising direction. The challenge resides in the ability to account for ground truth uncertainty in the evaluation of Causal ML methodologies. This could be done, for instance, by developing new evaluation metrics or analysis frameworks.

Another research direction consists of designing new experimental procedures to go beyond the simple measurement of causal effects made by RCTs and move toward counterfactual measures. In other words, experimental procedures need to be refined to reduce the current randomization and aggregation to approach counterfactual outcomes and provide partial counterfactual ground truths.

## 7. Conclusion

This work addresses the critical barriers that hinder the adoption and practical utility of causal machine learning methods, emphasizing the need for rigorous and systematic evaluation practices. By critically examining current evaluation limitations and their consequences, we argue that synthetic experiments, often criticized, are indispensable for understanding and assessing the capabilities of causal machine learning methods. To this end, we propose principles for pragmatic synthetic evaluation that prioritize transparency, comprehensive analyses, and standardization, fostering trust and reliability in causal machine learning research. While various challenges remain, including resource constraints and the intrinsic limitations of synthetic data to model unknown unknowns, the proposed framework lays out a foundation for advancing best practices and enabling the responsible deployment of causal machine learning methods across diverse, real-world applications. We hope that this work will encourage the Causal ML community to give more consideration to empirical evaluation and will inspire debate and research aimed at addressing the outlined limitations and refining the general principles proposed here to foster the adoption of Causal ML methods.

## Impact Statement

This work addresses a critical barrier to the broader adoption and responsible application of causal machine learning methods: the lack of rigorous and systematic evaluation practices. While causal methods hold immense potential to revolutionize decision-making by integrating predictive models with causal inference, unreliable evaluations hinder their practical utility. For example, in healthcare, poorly assessed models can lead to unsafe or inequitable treatment recommendations; in policymaking, they can unintentionally perpetuate discrimination or exacerbate societal inequalities.

This work provides a foundation for enhancing trust in causal machine learning by promoting more comprehensive and reliable evaluation practices. Beyond mitigating risks in sensitive applications, it establishes a pathway for ethical deployment and broader adoption of these methods, inspiring both collaboration and innovation within the field. This shift has the potential to shape how causal models are assessed, trusted, and deployed across diverse real-world settings.

## Acknowledgements

This research was partially funded by the French National Research Agency (ANR) under the France 2030 program, under the reference 23-PEIA-004, by the Artificial Intelligence for Safe and Smart Mobility Chair (Grant No. ANR-

23-CPJ1-0099-01), and by the UKRI Centre for Doctoral Training in Accountable, Responsible and Transparent AI (ART-AI) [EP/S023437/1]. We thank Elias Bareinboim for the enriching discussions and the anonymous reviewers for their thoughtful comments.

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

# A. Fundamentals on Causal Inference

## A.1. Identification Versus Estimation

A causal inference tasks have two distinct phases: **identification** that transforms causal queries into statistical ones by relying on theoretical assumptions and **estimation** that uses finite data and statistical models to approximate the transformed causal queries. **Identification** consists in showing whether a causal query is identifiable. Formally, given a class of causal models $\mathbb{M}$ (see Appendix A.2 for examples), an $\mathcal{L}_i$-query $Q(\mathcal{M})$ of a model $\mathcal{M} \in \mathbb{M}$ is identifiable from $\mathcal{L}_j$-data, $1 \leq j < i$, if for any pair of models $\mathcal{M}_1$ and $\mathcal{M}_2$ from $\mathbb{M}$, $Q(\mathcal{M}_1) = Q(\mathcal{M}_2)$ whenever $\mathcal{M}_1$ and $\mathcal{M}_2$ match in all $\mathcal{L}_j$ queries (Pearl, 2009). In other words, a causal query is said to be identifiable if, under an appropriate set of assumptions, it can be answered from lower-level data. Hence, the answer to the query exists and is unique. Once a query has been proven to be identifiable, it can be expressed in terms of statistical terms that can be estimated by combining statistical algorithms. The choice and application of statistical algorithms comprised the **estimation** phase.

While identification ensures a method is theoretically sound, it does not guarantee robustness in real-world settings, where assumptions are frequently violated or unfalsifiable (i.e., impossible to verify using only observable data (Bareinboim et al., 2022)). Therefore, it is essential to assess the ability of causal inference methods to approximate causal queries under practical constraints.

## A.2. Mathematical Tools to Answer Causal Questions

Among the mathematical frameworks used in causal inference, Structural Causal Models (SCMs) provide a representation allowing reasoning on the three levels of the PCH. A **Structural Causal Model** (Pearl, 2009) is a tuple $\mathcal{M} := \{\mathbf{V}, \mathbf{U}, \mathcal{F}, P(\mathbf{U})\}$. $\mathbf{V}$ constitutes the set of endogenous variables $\{V_1, V_2, ..., V_d\}$. They are the modeled variables. $\mathbf{U}$ is a set of exogenous variables $\{U_1, U_2, ..., U_l\}$. These variables represent the un-modelled causes of the endogenous variables. Hence, each endogenous variable $V_i$ is fully determined by a subset of variables in $\mathbf{V} \cup \mathbf{U}$, denoted $\boldsymbol{PA}(V_i) \cup \mathbf{U}_i$ where $\boldsymbol{PA}(V_i) \subseteq \mathbf{V} \backslash V_i$ are the parents of $V_i$ and $\mathbf{U}_i \subset \mathbf{U}$ are the exogenous (or latent) causes of $V_i$. $\mathcal{F}$ is a set of functions $\{f_1, f_2, \ldots, f_d\}$ called structural equations or causal mechanisms. Each function $f_i$ computes the variable $V_i$ from its parents and latent causes $\boldsymbol{PA}(V_i) \cup \mathbf{U}_i$. $P(\mathbf{U})$ is a probability function over the exogenous variables $\mathbf{U}$. An SCM characterizes a unique distribution over its endogenous variables, $P_{\mathcal{M}}(\boldsymbol{V})$, called the entailed distribution. The **causal graph** $\mathcal{G}$ of an SCM represents the cause-effect relationships between the variables. $\mathcal{G}$'s nodes are the SCM's variables $\mathbf{V} \cup \mathbf{U}$, and a directed edge $A \rightarrow V_i$ belongs to $\mathcal{G}$ whenever $A$ is a direct cause of $V_i$ $A \in \boldsymbol{PA}(V_i) \cup \mathbf{U}_i$ . From an SCM $\mathcal{M}$, one can derive any intervention and counterfactual distributions.

An **intervention** $\boldsymbol{do}(V_k = v)$ represents the action of setting the variable $V_k$ to the value $v$ in an arbitrary way (i.e., independently of any other variable) (Pearl, 2009). By manipulating the SCM $\mathcal{M}$ and defining a new SCM $\mathcal{M}_{\boldsymbol{do}(V_k=v)}$, changing the causal mechanism of $V_k$ to $f_k := v$, the intervention distributions $P_{\mathcal{M}}(\mathbf{V}|\boldsymbol{do}(V_k = v))$ can be computed as the entailed distribution of $\mathcal{M}_{\boldsymbol{do}(V_k=v)}$, $P_{\mathcal{M}_{\boldsymbol{do}(V_k=v)}}(\mathbf{V})$. Using intervention distributions, one can create causal measures summarizing information about interventions. For instance, the Average Treatment Effect (ATE) of the variable $T$ when intervened to the value $t$ instead of $c$ on another variable $Y$ is defined as the expectation of the difference between the intervention distributions $P_{\mathcal{M}}(Y|\boldsymbol{do}(T = t))$ and $P_{\mathcal{M}}(Y|\boldsymbol{do}(T = c))$, i.e., $\mathbb{E}[Y|\boldsymbol{do}(T = t)] - \mathbb{E}[Y|\boldsymbol{do}(T = c)]$ (resp. $\mathbb{E}[Y|\boldsymbol{do}(T = t), \mathbf{X} = \mathbf{x}] - \mathbb{E}[Y|\boldsymbol{do}(T = c), \mathbf{X} = \mathbf{x}]$). Another commonly used query in the literature is the CATE. The CATE of the variable $T$ when intervened to the value $t$ instead of $c$ on another variable $Y$ given that other variables $\mathbf{X}$ equal $\mathbf{x}$ is defined as the expectation of the difference between the intervention distributions $P_{\mathcal{M}}(Y|\boldsymbol{do}(T = t), \mathbf{X} = \mathbf{x})$ and $P_{\mathcal{M}}(Y|\boldsymbol{do}(T = c), \mathbf{X} = \mathbf{x})$, i.e., CATE $= \mathbb{E}[Y|\boldsymbol{do}(T = t), \mathbf{X} = \mathbf{x}] - \mathbb{E}[Y|\boldsymbol{do}(T = c), \mathbf{X} = \mathbf{x}]$.

A **counterfactual** consists of reasoning about the effect of an intervention in a context described by a factual realization (Pearl, 2009). Formally, given a factual realization $\mathbf{V}_F = \mathbf{v}_f$ and an intervention $\boldsymbol{do}(V_k = v)$, the counterfactual distribution $P_{\mathcal{M}}^{\mathbf{V}_F = \mathbf{v}_f}(\mathbf{V}|\boldsymbol{do}(V_k = v))$ correspond to the entailed distribution of the SCM $\mathcal{M}_{\boldsymbol{do}(V_k=v)}^{\mathbf{V}_F = \mathbf{v}_f}$ whose exogenous distribution equals the exogenous distribution of $\mathcal{M}$ given the factual realization $\mathbf{V}_F = \mathbf{v}_f$, i.e., $P_{\mathcal{M}_{\boldsymbol{do}(V_k=v)}^{\mathbf{v}_F=\mathbf{v}_f}}(\mathbf{U}) = P_{\mathcal{M}}(\mathbf{U}|\mathbf{V}_F = \mathbf{v}_f)$, and whose causal mechanism of the variable $\mathbf{V}_k$ has been modified to $f_k := v$.

Another commonly used class of causal models are Causal Graphical Models (CGMs). Unlike SCMs, CGMs only allow

reasoning up to the second level of the PCH. A **Causal Graphical Model** (Peters et al., 2017) is a tuple $\mathcal{M} := \{\mathbf{V}, \mathcal{P}\}$ where $\mathbf{V} = \{V_1, V_2, ..., V_d\}$ constitutes the set of endogenous variables and $\mathcal{P}$ is a set of distributions $\{p_1, p_2, \ldots, p_d\}$ called conditionals. Each distribution $p_i$ computes the probability of the variable $V_i$ given its parents $\boldsymbol{PA}(V_i)$. A CGM also induces a causal graph and an entailed distribution from which interventions can be computed similarly to SCMs.

## B. RealCause Experiment Details

### B.1. RealCause ATE Bias Over 100 Realizations and 20 Seeds

The following table presents the results of training RealCause on 100 different IHDP realizations, each evaluated across 20 random seeds. For each combination of realization and seed, the Average Treatment Effect (ATE) bias and the true ATE are recorded to illustrate the variability in performance.

| Realization Index | ATE Bias ($\mu \pm \sigma$) | True ATE |
|:---:|:---:|:---:|
| 1 | $0.377 \pm 0.216$ | 4.051 |
| 2 | $0.296 \pm 0.206$ | 4.099 |
| 3 | $0.373 \pm 0.352$ | 4.274 |
| 4 | $0.583 \pm 0.332$ | 4.162 |
| 5 | $0.395 \pm 0.247$ | 4.004 |
| 6 | $0.262 \pm 0.244$ | 3.991 |
| 7 | $0.364 \pm 0.277$ | 3.854 |
| 8 | $2.533 \pm 3.390$ | 10.466 |
| 9 | $1.483 \pm 1.650$ | 4.586 |
| 10 | $0.451 \pm 0.416$ | 3.948 |
| 11 | $0.704 \pm 0.633$ | 4.161 |
| 12 | $4.194 \pm 3.085$ | 12.649 |
| 13 | $0.337 \pm 0.211$ | 3.861 |
| 14 | $0.636 \pm 0.358$ | 3.908 |
| 15 | $0.384 \pm 0.258$ | 4.533 |
| 16 | $0.787 \pm 0.501$ | 3.637 |
| 17 | $0.286 \pm 0.227$ | 3.769 |
| 18 | $0.422 \pm 0.330$ | 3.941 |
| 19 | $0.337 \pm 0.300$ | 4.236 |
| 20 | $1.252 \pm 0.861$ | 3.155 |
| 21 | $0.443 \pm 0.249$ | 4.087 |
| 22 | $0.468 \pm 0.275$ | 4.313 |
| 23 | $0.512 \pm 0.459$ | 3.835 |
| 24 | $0.739 \pm 0.450$ | 4.596 |
| 25 | $2.683 \pm 1.742$ | 4.413 |
| 26 | $0.503 \pm 0.239$ | 3.756 |
| 27 | $5.298 \pm 5.206$ | 10.470 |
| 28 | $0.566 \pm 0.426$ | 3.745 |
| 29 | $0.577 \pm 0.541$ | 4.385 |
| 30 | $0.422 \pm 0.340$ | 4.336 |
| 31 | $0.491 \pm 0.235$ | 4.023 |
| 32 | $0.572 \pm 0.404$ | 4.157 |
| 33 | $4.315 \pm 2.810$ | 6.789 |
| 34 | $0.436 \pm 0.407$ | 3.756 |
| 35 | $0.607 \pm 0.464$ | 4.882 |
| 36 | $1.235 \pm 1.645$ | 3.343 |
| 37 | $0.613 \pm 0.447$ | 4.043 |
| 38 | $1.298 \pm 1.240$ | 6.221 |
| 39 | $0.547 \pm 0.667$ | 4.390 |

| Realization Index | ATE Bias ($\mu \pm \sigma$) | True ATE |
|---|---|---|
| 40 | $0.461 \pm 0.260$ | 4.230 |
| 41 | $0.356 \pm 0.279$ | 3.994 |
| 42 | $0.457 \pm 0.342$ | 4.499 |
| 43 | $0.771 \pm 0.534$ | 3.781 |
| 44 | $0.671 \pm 0.702$ | 3.299 |
| 45 | $0.966 \pm 0.863$ | 4.827 |
| 46 | $0.872 \pm 0.901$ | 5.066 |
| 47 | $0.350 \pm 0.256$ | 3.842 |
| 48 | $0.656 \pm 0.315$ | 4.395 |
| 49 | $0.575 \pm 0.551$ | 3.905 |
| 50 | $0.504 \pm 0.344$ | 4.306 |
| 51 | $0.412 \pm 0.275$ | 4.351 |
| 52 | $1.863 \pm 1.232$ | 5.791 |
| 53 | $0.891 \pm 0.741$ | 4.432 |
| 54 | $0.430 \pm 0.248$ | 3.740 |
| 55 | $0.333 \pm 0.321$ | 4.031 |
| 56 | $0.458 \pm 0.286$ | 4.085 |
| 57 | $0.452 \pm 0.496$ | 4.524 |
| 58 | $0.302 \pm 0.329$ | 4.242 |
| 59 | $1.055 \pm 1.049$ | 4.015 |
| 60 | $0.315 \pm 0.236$ | 4.082 |
| 61 | $0.396 \pm 0.185$ | 3.901 |
| 62 | $0.443 \pm 0.211$ | 4.480 |
| 63 | $0.450 \pm 0.382$ | 3.262 |
| 64 | $0.697 \pm 0.490$ | 4.177 |
| 65 | $0.353 \pm 0.350$ | 4.198 |
| 66 | $0.454 \pm 0.340$ | 4.051 |
| 67 | $2.595 \pm 2.189$ | 9.513 |
| 68 | $0.328 \pm 0.240$ | 3.814 |
| 69 | $0.300 \pm 0.183$ | 4.019 |
| 70 | $0.957 \pm 0.782$ | 5.857 |
| 71 | $0.382 \pm 0.229$ | 3.998 |
| 72 | $0.323 \pm 0.285$ | 3.838 |
| 73 | $0.325 \pm 0.232$ | 3.997 |
| 74 | $0.261 \pm 0.172$ | 3.996 |
| 75 | $0.854 \pm 0.724$ | 5.159 |
| 76 | $0.189 \pm 0.187$ | 4.147 |
| 77 | $0.422 \pm 0.201$ | 4.018 |
| 78 | $0.289 \pm 0.170$ | 4.196 |
| 79 | $0.462 \pm 0.243$ | 3.971 |
| 80 | $2.813 \pm 2.990$ | 9.524 |
| 81 | $1.187 \pm 0.935$ | 2.972 |
| 82 | $0.725 \pm 0.580$ | 4.152 |
| 83 | $1.533 \pm 1.356$ | 2.554 |
| 84 | $6.209 \pm 11.319$ | -0.604 |
| 85 | $2.194 \pm 1.979$ | 5.888 |
| 86 | $0.437 \pm 0.380$ | 4.912 |
| 87 | $0.264 \pm 0.209$ | 3.983 |
| 88 | $0.602 \pm 0.427$ | 3.946 |
| 89 | $0.532 \pm 0.379$ | 3.945 |
| 90 | $0.329 \pm 0.262$ | 3.917 |

| Realization Index | ATE Bias ($\mu \pm \sigma$) | True ATE |
|---|---|---|
| 91 | $0.403 \pm 0.280$ | 3.570 |
| 92 | $3.097 \pm 3.135$ | 8.456 |
| 93 | $0.506 \pm 0.408$ | 4.239 |
| 94 | $0.443 \pm 0.349$ | 3.850 |
| 95 | $0.702 \pm 0.612$ | 4.087 |
| 96 | $0.508 \pm 0.339$ | 4.355 |
| 97 | $1.374 \pm 0.917$ | 6.806 |
| 98 | $0.425 \pm 0.415$ | 4.134 |
| 99 | $0.349 \pm 0.350$ | 4.032 |
| 100 | $0.386 \pm 0.251$ | 4.016 |

### B.2. Realism Checks

RealCause provides post hoc realism checks, such as Kolmogorov-Smirnov (KS) tests, which can help guide model selection. However, these checks are neither enforced nor required. Poor performance may coincide with failed realism tests. To evaluate the effectiveness of these realism checks in improving model performance, we compare the distribution of ATE bias across all seeds to that of the subset passing realism checks.

Specifically, we use the following approach:

- We apply 100 univariate two-sample KS tests separately to treatment and outcome variables, comparing the generated data's marginal distributions to those of the real data. This results in 100 p-values for each variable (treatment and outcome).

- We aggregate these p-values and consider a model to pass the realism check if at least 70% of them exceed a 0.05 threshold, indicating no significant difference between generated and real data distributions (as done in RealCause).

- For each IHDP dataset realization, we train 20 models with different random seeds and compare the full set of ATE biases to the subset corresponding to models that pass the realism check.

Next, we performed Welch's t-test to assess whether the realism filter led to statistically significant differences in ATE bias. Our results showed that only 1 out of 100 realizations exhibited a statistically significant difference (at a 0.05 significance level). This suggests that realism checks rarely identify subsets with meaningfully lower ATE bias.

We emphasize that our goal is not to explain the source of ATE errors. Instead, we aim to highlight that poor realism often correlates with high ATE error, but the reverse is not guaranteed. This behavior aligns with the Causal Hierarchy Theorem, which states that agreement at the level of observational distributions does not imply agreement on intervention distributions. Therefore, filtering models based on realism—i.e., fit to observational data—may not consistently lead to improved ATE estimation.

## C. CausalNF Experiment Details

The Triangle$_{\text{NLIN}}$ (Javaloy et al., 2023) SCM is defined as follows:

$$
\begin{aligned}
x_1 &= f_1(u_1) = g(u_1) + 1, \\
x_2 &= f_2(x_1, u_2) = 2x_1^2 + g(u_2), \\
x_3 &= f_3(x_1, x_2, u_3) = \frac{20}{1 + e^{-x_2^2 + x_1}} + g(u_3),
\end{aligned}
$$

where $u_i \sim N(0, 1)$ and $g(u_i)$ is the identity mapping in the original formulation. For the experiments of Section 3.2, we introduce two modifications to $g(u_i)$ to violate diffeomorphic assumptions of the mechanisms:

1. *Segmented Linear Function:*

$$g(U) = \begin{cases} U & \text{if } U < c_1, \\ c_1 & \text{if } c_1 \leq U < c_2, \\ U - (c_2 - c_1) & \text{if } U \geq c_2. \end{cases}$$

This creates a non-bijective region within the interval $[c_1, c_2]$, violating bijectivity and introducing non-differentiability at the interval boundaries.

2. *Sinusoidal Function:*

$$g(U) = \sin(2\pi f U),$$

where $f$ is a frequency. This function keeps the mechanism differentiable but is entirely non-bijective, with periodic overlaps that map multiple noise values to the same output.

Table 2 shows the RMSE performance across different configurations of $g(U)$ and parameter settings. We hypothesize that these results stem from the specific structure of the tested SCM, where the noise variables do not directly interact with the causal parents (non-bijective collisions have little additive effect on the outcome). For example, although the segmented linear function introduces regions where multiple $u_i$ values map to the same $g(u_i)$, selecting an incorrect $u_i$ has no impact on performance. This is because the transformation maps back to the same $g(u_i)$, and the value of $u_i$ does not otherwise influence the functions. Moreover, the non-bijectivity of the noise transformation might actually simplify the learning process by reducing the number of distinct mappings that need to be modeled.

*Table 2.* Performance of CausalNF under various modifications to the noise function $g(U)$. *SEG-LINEAR-0.2* represents a segmented linear function with a non-bijective interval of length 0.2 centered at 0.5, while *SINUSOID-0.25* represents a sinusoidal function with a frequency of 0.25. The table reports the mean and standard deviation of the RMSE for counterfactual estimation. Results averaged over five runs.

| SCM Variant | RMSE CF (Mean $\pm$ Std. Dev) |
|---|---|
| TRIANGLE$_{NLIN}$ | $0.127 \pm 0.021$ |
| SEG-LINEAR-0.2 | $0.132 \pm 0.026$ |
| SEG-LINEAR-0.4 | $0.115 \pm 0.014$ |
| SEG-LINEAR-0.6 | $0.108 \pm 0.016$ |
| SEG-LINEAR-0.8 | $0.092 \pm 0.009$ |
| SEG-LINEAR-1.0 | $0.083 \pm 0.012$ |
| SINUSOID-0.25 | $0.061 \pm 0.015$ |
| SINUSOID-0.30 | $0.063 \pm 0.012$ |
| SINUSOID-0.35 | $0.060 \pm 0.008$ |
| SINUSOID-0.40 | $0.062 \pm 0.007$ |
| SINUSOID-0.45 | $0.067 \pm 0.011$ |
| SINUSOID-0.5 | $0.069 \pm 0.014$ |
| SINUSOID-1.0 | $0.062 \pm 0.006$ |
| SINUSOID-2.0 | $0.081 \pm 0.013$ |
| SINUSOID-4.0 | $0.078 \pm 0.010$ |
| SINUSOID-6.0 | $0.066 \pm 0.011$ |
| SINUSOID-8.0 | $0.064 \pm 0.011$ |
| SINUSOID-10.0 | $0.085 \pm 0.020$ |

**Details on the Non-Identifiable Counterfactual Experiment**

We adopt a counterexample from Nasr-Esfahany & Kiciman (2023), in which two structurally distinct SCMs (denoted as CTF1 and CTF2) share the same observational distribution but differ in their counterfactual distributions, meaning that the counterfactual queries are not identifiable. The adapted SCMs form a simple chain structure and are defined as:

*Table 3.* Performance of CausalNF on chain SCMs with differing $f_2$ functions. Results averaged over five runs.

| SCM | RMSE CF (Mean, $\pm$ Std. Dev.) |
|---|---|
| CHAIN[CTF1] | $0.158 \pm 0.009$ |
| CHAIN[CTF2] | $0.583 \pm 0.011$ |

$$x_1 = u_1,$$
$$x_2 = f_2(x_1, u_2),$$
$$x_3 = f_3(x_2, u_3),$$

where $f_3(x_2, u_3) = x_2 + u_3$ is fixed for both SCMs, and $f_2$ differs between two variants:

$$f_2(x_1, u_2) = \begin{cases} u_2 & \text{if } x_1 \geq 0, \\ u_2 - 1 & \text{if } x_1 < 0, \end{cases} \quad \text{(defining CTF1),}$$

or

$$f_2(x_1, u_2) = \begin{cases} u_2 & \text{if } x_1 \geq 0, \\ -u_2 & \text{if } x_1 < 0. \end{cases} \quad \text{(defining CTF2).}$$

The function $f_2$ is diffeomorphic everywhere except at $x_1 = 0$, where it is discontinuous and not invertible. However, it remains invertible with respect to $u_2$.

$$\text{CTF1: } x_1 = u_1,$$
$$x_2 = \begin{cases} u_2, & \text{if } x_1 \geq 0, \\ u_2 - 1, & \text{if } x_1 < 0, \end{cases}$$
$$x_3 = u_3 + x_2.$$

$$\text{CTF2: } x_1 = u_1,$$
$$x_2 = \begin{cases} u_2, & \text{if } x_1 \geq 0, \\ -u_2, & \text{if } x_1 < 0, \end{cases}$$
$$x_3 = u_3 + x_2.$$

Training CausalNF on these two SCMs reveals a stark contrast in performance. As summarized in Table 3, the RMSE for counterfactual estimation is significantly higher for CTF2 than for CTF1, suggesting that CausalNF consistently defaults to learning the structure of CTF1 regardless of the true underlying SCM.

We extend this experiment by modifying $f_3$ to introduce a similar noise-parent interaction. Specifically:

$$f_3(x_2, u_3) = \begin{cases} u_3 & \text{if } x_2 \geq 0, \\ u_3 - 1 & \text{if } x_2 < 0, \end{cases}$$

and

$$f_3(x_2, u_3) = \begin{cases} u_3 & \text{if } x_2 \geq 0, \\ -u_3 & \text{if } x_2 < 0. \end{cases}$$

resulting in two new SCMs: CTF3 and CTF4 in which $f_2$ and $f_3$ have similar structure.

*Table 4.* Performance of CausalNF on chain SCMs with additional noise-parent interaction in $f_3$. Results averaged over five runs.

| SCM | RMSE CF (Mean $\pm$ Std. Dev.) |
|---|---|
| CHAIN[CTF3] | $0.147 \pm 0.010$ |
| CHAIN[CTF4] | $1.037 \pm 0.013$ |

CTF3: $x_1 = u_1,$

$$x_2 = \begin{cases} u_2, & \text{if } x_1 \geq 0, \\ u_2 - 1, & \text{if } x_1 < 0, \end{cases}$$

$$x_3 = \begin{cases} u_3, & \text{if } x_2 \geq 0, \\ u_3 - 1, & \text{if } x_2 < 0. \end{cases}$$

CTF4: $x_1 = u_1,$

$$x_2 = \begin{cases} u_2, & \text{if } x_1 \geq 0, \\ -u_2, & \text{if } x_1 < 0, \end{cases}$$

$$x_3 = \begin{cases} u_3, & \text{if } x_2 \geq 0, \\ -u_3, & \text{if } x_2 < 0. \end{cases}$$

The results, shown in Table 4, demonstrate even more pronounced failure for CTF4, where CausalNF erroneously models $f_2$ and $f_3$ both as the simpler $f_2$ variant of CTF1, doubling the error.

## D. Extending the CausalNF Experiments

Say one wants to extend the experiments carried out in Section 3.2 on the robustness of CausalNF (Javaloy et al., 2023) against violation of the bijectivity assumption. One could, for example, define the following synthetic experimental design choices:

i) **Set of causal models:** The SCMs $\mathcal{M} = \{\mathcal{M} := \{\mathcal{G}_{\mathcal{M},\mathcal{F},P(\mathbf{U})}\}\}$ such that $\mathcal{G}_{\mathcal{M}}$ is a DAG, $\mathcal{F}$ is a set of fully-connected neural networks and $P(\mathbf{U}) \sim \mathcal{U}([0,1])$. $\mathcal{G}_{\mathcal{M}}$ is parametrized by its size $|\mathbf{V}|$ (i.e., the number of endogenous variables) and $\mathcal{F}$ by its depths, numbers of neurons and activation functions.

ii) **Set of queries:** Counterfactuals over $\mathcal{M}$ given the dataset $\mathcal{D}$, $\mathcal{Q}_{\mathcal{M},\mathcal{D}} = \{P_{\mathcal{M}}^{\mathbf{V}=\mathbf{v}_f}(Y|\boldsymbol{do}(V_k = v_k)) \mid Y \in \mathbf{V}, V_k \in \mathbf{V} \setminus Y, \mathbf{v}_f \in \mathcal{D}, v_k \in \mathcal{D}_k\}$ where $\mathcal{D}_k$ is the realizations of $V_k$ in the dataset $\mathcal{D}$

iii) **Set of training data:** Datasets of size $N$ drawn from the entailed distribution of $\mathcal{M}$ without perturbation, $\mathcal{D}_{\mathcal{M}} = \{\mathcal{D} = \{\mathbf{v}_i\}_{i=0}^{N} \sim P_{\mathcal{M}}(\mathbf{V})\}$

iv) **Generation algorithm:**

    1- uniform sampling over the ranges of all the control parameters (i.e., $\mathcal{G}_{\mathcal{M}}$ size, $\mathcal{F}$ depths, numbers of neurons and activation functions, dataset sizes)

    2- uniform sampling of a DAG of size $|\mathbf{V}|$ (Talvitie et al., 2020)

    3- initialization of the neural networks weights and bias following the Glorot uniform (Glorot & Bengio, 2010)

    4- dataset creation from sampling $N$ observations from $\mathcal{M}$

    5- query creation by randomly picking the variables $Y$ and $V_k$ from the sets $\mathbf{V}$ and $\mathbf{V} \setminus Y$

    6- randomly sampling from the training set $\mathbf{v}_f$ and $v_k$ as realizations of $\mathbf{V}$ and $V_k$

v) **Induced distribution:** The joint distribution of the control parameters of the SCMs, queries, and datasets is, by definition, multivariate uniform with independent components. However, the distribution of observable characteristics,

such as the bijectivity of the causal mechanisms or the existence of interaction between endogenous and exogenous variables, is not straightforward to determine. Hence, for the characteristics that may be relevant to consider when studying the behavior of CausalNF when the bijectivity assumption is violated, it will be necessary to carry out an empirical analysis of their distribution by analyzing the SCMs sampled by the generation algorithm.

