# OpenReview forum: "Position: Causal Machine Learning Requires Rigorous Synthetic Experiments for Broader Adoption"
_ICML.cc/2025/Position_Paper_Track — ICML 2025 Position Paper Track poster_

### Official Review · Reviewer_fVML · 2025-02-15

**Significance:** 4
**Argument Clarity:** 4
**Rating:** 4
**Confidence:** 3

**Questions:**

•	Should the initial causal assumptions, such as ignorability, independence, and diffeomorphism, also be explicitly stated in advance, in addition to the five elements that need to be clarified in synthetic experiments in Section 4.2?

•	Regarding Sections 4.2 and 4.3, are there any open-source repositories that enable researchers to specify any causal DAG structures along with generation algorithms to model causal relationships and generate synthetic data accordingly? If not, this could be a promising direction for future research.

**Discussion Potential:**

4

**Paper Summary:**

The paper presents a position that synthetic experiments are important to precisely understand and assess the capabilities of causal ML methods. The authors point out that existing empirical evaluation of causal ML methods suffer from several problems, such as scarcity of ground truth data, inherent bias or non-identifiability in (semi-)synthetic data, and oversimplicity of synthetic experiments. The authors then demonstrate these problems by conducting experiments using existing causal ML methods such as RealCause and CausalNF. Based on these problems, the authors suggest four principles to mitigate these problems and properly evaluate the causal ML methods on not only accuracy but also other important aspects such as robustness, identifiability, and interpretability.

**Position:**

Yes

**Position In Title:**

Yes

**Related Work:**

4

**Strengths And Weaknesses:**

**Strengths**:

•	As mentioned in the paper, the difficulty in evaluating causal ML frameworks due to lack of observations on counterfactual outcomes is a major pain point in the field of causal ML and worths further exploration.

•	The authors offer a thorough discussion on the challenges of empirically evaluating causal ML methods, supported by sufficient references from the existing literature and demonstrations with experiments.

•	The paper clearly outlines the current limitations of synthetic datasets in research and introduces 4 principles to address these challenges with sufficient detail.

**Weaknesses**:

•	The authors could also provide some examples of developing standardized evaluation frameworks for causal ML methods. For example, what evaluation metrics shall we include in these evaluation frameworks?

•	I somewhat disagree with authors’ opinion that synthetic experiments cannot capture unknown unknowns. For example, in spatial causal inference, researchers make efforts in modeling unobserved confounding variables in the synthetic datasets [1].

References:

[1]. Reich, B. J., Yang, S., Guan, Y., Giffin, A. B., Miller, M. J., & Rappold, A. (2021). A review of spatial causal inference methods for environmental and epidemiological applications. International Statistical Review, 89(3), 605-634.

**Support:**

4

---

> ### Author Rebuttal · Authors · 2025-03-31
>
> We thank the reviewer for the positive assessment and suggestions on how to further improve our proposal. Below, we address their concerns.
>
> **Q1: "Should the assumptions be explicitly stated in advance?"**: The causal assumptions can be stated in advance, hence restricting the domain of considered experiments. However, they do not necessarily need to be explicitly stated in advance, especially when one considers a larger set of experiments (e.g., semi-Markovian SCMs) when some assumptions are not guaranteed to be respected (e.g., absence of hidden confounding). In such a case, the assumptions have to be tracked experiment per experiment to assess whether they are violated or not, hence resulting in a robustness analysis experiment or an analysis of the performance within the identification domain. In short, whether analyzed upstream or downstream, assumptions need to be monitored to assess their impact on the performance of Causal ML methods.
>
> **Q2: "Are there any open-source repositories to specify a DAG and generate synthetic data accordingly?"**: Unfortunately, to the best of our knowledge, there are no such open-source repositories, and we also think it would be a promising direction for future research. Through this paper, we hope to raise awareness of the Causal ML community on this topic and encourage the development of such solutions. To provide a balanced answer, we would like to point out that although no complete solution exists to date, some parts of the problem have already been studied.
>
> For instance, recent progress has been made in the sampling of DAG (Talvitie et al., 2020; Harviainen & Koivisto, 2024).
> Some open-source repositories also generate SCMs and synthetic data randomly (under specific assumptions, such as the absence of hidden confounding) but do not implement L2 or L3 query estimation as they have been designed for the evaluation of causal discovery methods (Kalainathan et al., 2020; Gupta et al., 2023). Moreover, these works fall short in their implementation of Principles 2 and 3. For instance, the distribution of the sampled SCMs is not studied, and no guarantees, such as strict positivity of the SCMs distribution, are given either.
>
> In addition, there also exist open-source tools that enable the simulation of synthetic SCMs and estimate ground truth queries (DoWhy, Dagitty, Fusion, PyMC). However, the SCMs must be manually designed, specifying the causal graph, exogenous variables distributions, and causal mechanisms.
>
> Although limited to the simulation of ATE, the work of Rudolph et al, 2023 seems to be the closest to what you are asking for. Given a precise causal structure (triangular with covariates as common causes of the treatment and the outcome) and a set of input parameters, causal mechanisms are randomly generated and used to sample a synthetic dataset and estimate the ATE ground truth.
>
> **Weakness 1: "Missing examples like what evaluation metrics shall we include?"**: In order to encompass all Causal ML methods, we have decided to remain general in the guidelines and principles proposed. We recognize that a considerable amount of work remains to be done to detail these principles for each task or application case. That's why we think this position paper deserves to be read by the broader ML community (see answer to Question 1 of Reviewer Vkum). So, as you point out, the choice of metrics is a question not explored in this work because we consider it too task and application-dependent to give global guidelines.
>
> **Weakness 2: "Disagreement with the claim that synthetic experiments cannot capture unknown unknowns"**: We call an unknown unknown something one ignores that one does not know. In the example of hidden confounding you mention, modeling a hidden confounder would be considered a known unknown. It is something we know exists, even if we cannot measure it. It is, therefore, modeled and taken into account. In this example, the unknown unknown situation would be to ignore the existence of a second hidden confounder. Consequently, as it is not included in the simulation, the method is evaluated without its effect. In a real dataset, on the other hand, if such a hidden confounder exists, its effect is always present regardless of our awareness of its existence.

---

### Official Review · Reviewer_tNwk · 2025-03-09

**Significance:** 3
**Argument Clarity:** 4
**Rating:** 3
**Confidence:** 5

**Questions:**

- The main challenge might be that real world scenarious are quite complicated and it is difficult to capture this in an experiment. What would be your recomendations for this?
- Currently methods for sensitivity analysis in causal models are developed. Could this be an alternative to support the adoption of causal ML?

**Discussion Potential:**

3

**Paper Summary:**

The starting point of the paper is the observation that Causal ML modes are underutilized in industry and real world applications. The authors claim that this is due to lacking convincing evaluation of causal models. The authors argue that synthetic experiments are key to understand causal models and they propose principles how to evaluate causal models.

**Position:**

Yes

**Position In Title:**

Yes

**Related Work:**

3

**Strengths And Weaknesses:**

Strenghts:
- The paper is well structured and written and easy to follow.
- The paper takes up an important practical problem.

Waknesses:
- The derived recommendations / principles might be not sufficient enough.

**Support:**

4

---

> ### Author Rebuttal · Authors · 2025-03-31
>
> We thank the reviewer for the thoughtful review. Below, we respond to their questions and comments.
>
> **Q1: "How to capture real-world complexities in a synthetic experiment?"**: Thank you for this relevant question. Indeed, one can rapidly find oneself overwhelmed by the complexity of a simulation approaching reality.
>
> We recommend implementing a progressive increase in simulation complexity: starting with an intuitive experiment (e.g., small graph, simple mechanisms, binary variables, sufficient data, and no assumptions violation) and adding one new complexity at a time (e.g., presence of hidden confounders, selection bias, small data or mixture of SCMs). For each new complexity, one should analyze its impact on method performances before adding the next one. By doing so, one isolates the effect of previously added complexities while keeping experiments getting more and more complex. Among the possible complexities, one could also consider the presence of outliers, violation of ignorability assumption, erroneous causal graph (or adjustment set), etc.
>
> We suggest adding this recommendation to Principle 2 to help researchers and practitioners make explicit all the assumptions they make, including in such complex simulations.
>
> **Q2: "Could sensitivity analysis methods be an alternative to support the adoption of Causal ML?"**: Sensitivity analysis methods (Frauen et al., 2024, Schronder et al., 2023) as well as partial identification methods (Zaffalon et al., 2020; Zhang et al., 2022), that provide bounds on causal estimates (rather than point estimates) relaxing certain assumptions, are indeed important support for the wider adoption of Causal ML, particularly when sensitivity analysis is performed on unfalsifiable assumptions such as the presence of hidden confounders. However, they remain Causal ML methods that need to be rigorously evaluated to identify in which settings the bounds on causal estimates are accurately estimated.

---

### Official Review · Reviewer_qF3S · 2025-03-10

**Significance:** 2
**Argument Clarity:** 3
**Rating:** 3
**Confidence:** 3

**Questions:**

- Why did you only choose RealCause and not consider other generative algorithms?
- Did you evaluate the MSE of the estimators across different monte carlo iterations from RealCause?
- RealCause asks authors to ensure the generated data distribution is not different than the original empirical data distribution: did you do this? I did not see it reported anywhere, and this behavior could explain why RealCause's geneated ATEs are vastly different than the true ATE.

**Discussion Potential:**

1

**Paper Summary:**

The authors argue for more wide-spread adoptioon of fully synthetic experiments to benchmark causal machine learning estimators. They first highlight the main limitations with using current standards: RCTs are too expensive and are sometimes completely infeasible; semi-synthetic experiments are "untrustworthy" and can yield bias results; and current simulation setups are too simple for real world generalizability. The authors present a case-study highlighting how semi-synthetic designs can produce biased results with the RealCause algorithm and CausalNF estimator. The authors conclude with best-practices for fully synthetic experiments.

**Position:**

Yes

**Position In Title:**

Yes

**Related Work:**

3

**Strengths And Weaknesses:**

## Strengths

The authors effectively balance idealism and pragmatism. They acknowledge that alternatives to their position, specifically that RCTs are gold standard for treatment effect estimation and benchmarking. However, they highlight what makes them so unrealistic for widespread use: the infeasibility and cost.

Best practices for simulations are clearly highlighted and actionable by experts.

## Weaknesses
The authors use a single, older algorithm to highlight why current generative evaluation techniques are flawed. However, one non-SOTA method not working does not mean the entire state of generative methods for validation does not work. The same goes for choosing a base estimator: just because CausalNF is not stable across runs does not mean that the entire evaluation framework is flawed. The authors do not clearly argue or justify why the conclusion based on one approach and estimator can generalize to all distinct methods.

Further, I'm not actually convinced that RealCause does not work. Specifically, the authors claim that RealCause fails because "Relying on  a single seed conceals this variability [in method rankings], making benchmarks fragile and potentially misleading." However, no researcher should be evaluating an estimator's performance based on a single Monte Carlo iteration/seed. Further, researchers should not be evaluating causal ML methods based on _rankings_ but rather a summary across runs, eg MSE. I am not familiar with any papers in the field that evaluate based on rankings, but that evaluation metric is fundamentally flawed and not the experimental setup.

The authors also argue that RealCause produces noisy estimates because the true ATE across runs is highly different: that seems like a good thing. Most real data is noisy and variable and will produce varied ATEs across draws from the data generating process (this is why the field cares not only about point estimation but also about uncertainty quantification).

In lines 231-233, the authors argue that because RealCause cannot work for counterfactual estimation that generative methods should not be used in the field. However, counterfactual estimation is a very different type of causal query than ATE estimation which is a very different type of causal query than CATE estimation which is a very different type of causal query than LATE estimation. It is unclear why this is a bad thing.

I am not sure that the authors propose a _new_ or discussion-worthy position. The field currently relies on fully-synthetic experiments because of the author's recognized limitations. While the best practices may be a useful contribution, those best-practices do not defend the author's position that more rigorous experiments will lead to more widespread adoption.

## Post-rebuttal comments
I think I better follow the argument now. You want to highlight that there is no consensus as to which treatment effect estimator is best across many different RealCause-generated datasets. This experiment simply highlights that an analyst could run into such an issue with generative benchmarking methods and that such approaches are still flawed.

I think the extra paragraphs on "Impact on adoption" would be very helpful.

I think the authors have addressed most of my concerns. As such, I am raising my score from a weak reject to a weak accept.

**Support:**

2

---

> ### Author Rebuttal · Authors · 2025-04-01
>
> We thank the reviewer for their detailed feedback. We appreciate the opportunity to clarify our intent and better explain our findings.
>
> **Q1: "Why only RealCause?"**: We considered other works such as the ones of Manela et al., 2024; Athey et al., 2020 or Parikh et al., 2022. However, RealCause has already been used in several research works for evaluation of causal inference methods (Pros and Vitria, 2024; Piskorz et al., 2025; Bozorgi, 2021; Ter-Minassian et al., 2024; Shoush and Dumas, 2025; Mahajan et al., 2024; Zhang et al., 2024; van der Laan et al., 2024). Hence, we prioritized illustrating the potential limitations of this method.
>
> **Q2: "MSE of estimators across different MC iterations"**: We are not entirely sure what "MC iterations" means in this context, so we clarify how our experiments were structured. As IHDP is composed of several datasets, we call a realization one dataset of IHDP, which is associated with a specific ground truth ATE. We trained a new generative model using RealCause for each combination of realization and seed and reported ATE bias averaged over 20 seeds per realization. RealCause does not model the covariate distribution P(W) and relies on fixed, real W sampled from empirical datasets for ATE evaluation to avoid using unrealistic covariate combinations. We follow this protocol and evaluate ATE on real data using a 50/40/10 train/validation/test split rather than generating new W. Lastly, we run 100 univariate KS tests per model to assess realism.
>
> **Q3: "Realism tests when evaluating RealCause"**: Thank you for this observation. RealCause does not define formal acceptance criteria for selecting generative models. It provides post hoc realism checks (e.g., KS tests) that can help guide model selection. However, they are not enforced nor required. This is why we did not report them.
>
> However, we agree that poor performance might sometimes coincide with failed realism tests. To test this, we compared the distribution of ATE bias across all seeds to that of the subset passing realism checks (i.e., 70% of treatment and outcome KS tests with p-values > 0.05), using Welch's t-test per realization. Only 1 out of 100 realizations showed a statistically significant difference, indicating that realism checks rarely identify subsets with meaningfully lower ATE bias.
>
> Indeed, no matter how one searches for a seed that matches the observational distribution, the underlying generative model may simply lack the information needed for accurate ATE estimation. While poor realism may often correlate with high ATE error, as you suggested, the reverse is not guaranteed. This reflects the Causal Hierarchy Theorem, stating that observational distribution match does not imply agreement on intervention distributions. We note that our goal is not to explain the source of ATE errors. We further underline that when the ground-truth ATE is unknown, one should assess the capabilities of the method to properly approximate it (using synthetic experiments) before benchmarking another Causal ML method.
>
> **Weakness 1 "Overgeneralizing"**: Our goal is not to argue that all methods and evaluations are flawed nor to claim that RealCause or CausalNF are universally unreliable. Instead, we aim to demonstrate that evaluation issues can arise even in widely used setups and raise a broader issue regarding evaluation practices. This is not an overgeneralization but an existence proof. Our conclusions are intended to encourage better practices, not to generalize failure from a single case. We will clarify this further in the paper.
>
> **Weakness 2 "Rankings"**: Our use of the term "ranking" refers to the implicit comparisons made when new method papers claim that one method performs "better" than another one based on benchmark results on a given dataset or task. This still constitutes a form of ranking, even if it is not a proper leaderboard.
>
> **Weakness 3 "true ATE is highly different"**: Our paper does not discuss the true ATE or its effects. All our analyses refer to the ATE error, i.e., the deviation between the estimated ATE and the known ground-truth ATE. We mention true ATE once (line 186), and our point is that while the true ATE remains constant for a given realization, RealCause can yield highly variable ATE estimates across different seeds. This variance in error–not in the ground-truth effect–is what we highlight as a concern.
>
> **Weakness 4 "Generative methods should not be used & RealCause for counterfactual estimation"**: Lines 231–233 (second column) do not argue that generative methods should not be used, nor do they criticize RealCause for counterfactual estimation. Rather, the point made is that synthetic data is necessary when ground-truth counterfactuals are required for evaluation. If the line numbers referenced were inaccurate, we would be happy to clarify further.
>
> **Weakness 5 "those best practices do not defend the position"**: Please refer to Q1 and Q2 of Reviewer Vkum for clarification.

---

> > ### Comment · Reviewer_qF3S · 2025-04-03
> >
> > I thank the authors for their detailed response. I have a few questions and would appreciate some clarity. But I keep my sentiment that this paper would not lead to meaningful discussions within the field.
> > ---
> > - "We considered other works such as the ones of Manela et al., 2024; Athey et al., 2020 or Parikh et al., 2022"
> >   - Do you have experimental evaluation of these methods? I could not find such an evaluation
> > - We are not entirely sure what "MC iterations" means in this context
> >   - Sorry for the use of short hand. MC iterations means monte carlo iterations, which is essentially what you do by considering data drawn from multiple seeds.
> > - "To test this, we compared the distribution of ATE bias across all seeds to that of the subset passing realism checks (i.e., 70% of treatment and outcome KS tests with p-values > 0.05), using Welch's t-test per realization."
> >   - Can you clarify how you are using the KS test in this context? I would think you should use the KS test or some other test of distributional difference to see if the entire distribution of the generated data differs from the distribution of the original data. Do you mean that 70% of simulated datasets were not statistically significantly different?
> > - "This is not an overgeneralization but an existence proof. Our conclusions are intended to encourage better practices, not to generalize failure from a single case. "
> >   - Why do you need to prove that a failure case exists? I do not see how this claim offers support to the overall argument that we need more synthetic experiments

---

> > > ### Author Response · Authors · 2025-04-04
> > >
> > > We thank the reviewer for the follow-up and appreciate their engagement. Below are the requested clarifications.
> > >
> > > > Do you have experimental evaluation of Manela et al., Athey et al., or Parikh et al.?
> > >
> > > No, we do not present experiments on those methods. In our original response, we said that we explored several methods during scoping but ultimately focused on RealCause due to its broader adoption in recent literature and its availability for benchmarking.
> > >
> > > > Can you clarify how you are using the KS test in this context? Do you mean that 70% of simulated datasets were not statistically significantly different?
> > >
> > > Before clarifying how we used the KS tests (realism test), we would like to reiterate that due to the fundamental limitations established by the Causal Hierarchy Theorem (agreement at the level of observational distributions does not imply agreement at the level of intervention or counterfactual distributions). In other words, the realism tests are not an appropriate diagnostic for evaluating causal validity. Therefore, **filtering models based on realism (i.e., fit to observational data) and noticing that this does not consistently lead to improved ATE estimation is the expected behavior under the theorem, independent of the choice of realism test**. This is also why we don't report this in the paper.
> > >
> > > Regarding how we use the KS test:
> > > Following RealCause's methodology and codebase, we perform 100 univariate two-sample KS tests, applied separately to treatment and outcome variables. Each test compares the generated vs. real marginal distribution of a single variable, producing a set of 100 p-values per variable (treatment and outcome). We then aggregate these p-values (per variable) using summary statistics such as the mean p-value and the 30th percentile p-value ("70% of simulated datasets were not statistically significantly different from the real data distribution"). So far, we mirror RealCause's methodology and code.
> > >
> > > We apply 3 filtering strategies using combinations of the above 2 statistics. We consider a model to pass the realism test if the selected summary p-values exceed 0.05. For each dataset realization, we train 20 models with different seeds, allowing us to compare:
> > > 1. The full set of ATE biases from all 20 models.
> > > 2. The subset of ATE biases from models that pass the chosen realism check (smaller, but non-empty).
> > >
> > > We then perform Welch's t-test to assess whether the realism filter yields a statistically significant difference in ATE bias. Across all 3 strategies, only 1 out of 100 realizations showed a statistically significant difference (at a 0.05 significance level, so 1/100 is within expected noise).
> > >
> > > > Why do you need to prove that a failure case exists? How does this claim offer support to the overall argument that we need more synthetic experiments?
> > >
> > > Existence proofs are a powerful rhetorical and scientific device to highlight overlooked/under-discussed limitations, making them well-suited for a position paper.
> > > Showing that failure cases are real in prominent and widely known methods is sufficient motivation to advocate for more rigorous evaluation standards across the field.
> > > We are not claiming that all generative evaluations are flawed, nor that RealCause or CausalNF are fundamentally unusable. Instead, our experiments highlight how fragile or biased results can arise even in respected, widely used pipelines.
> > >
> > > This supports our broader argument: the field needs more systematic synthetic experimentation because real-world validation is difficult. For example, agreement on observational metrics is insufficient for causal guarantees. This is why causal ground truths need to be used for evaluation. As such, real-world ground truths are scarce or even impossible to get in some settings (cf. Problem 1), so we argue that synthetic experiments are necessary to also cover these settings. Our experiments help make this concrete, revealing subtle but important pitfalls that can go undetected without rigorous synthetic experiments.
> > >
> > > > Feedback incorporated
> > >
> > > We thank the reviewer for the feedback and questions, which we feel will help us improve the paper.
> > >
> > > We take note of the critiques:
> > >
> > > (a) Lack of explanation of why we chose to study RealCause
> > >
> > > (b) Unclear explanation of the link between this experiment and the paper's position
> > >
> > > (c) Lack of argumentation highlighting how rigorous synthetic experiments applying the suggested best practices would allow broader adoption of Causal ML methods
> > >
> > > We will revise the paper as follows:
> > >
> > > (i) Explain in Section 3.1 that we chose RealCause because it is already broadly used in the literature
> > >
> > > (ii) In Section 3.1, clarify the goal of this experiment and how it supports our position
> > >
> > > (iii) Clarify how the proposed best practices can lead to broader adoption of Causal ML methods in the broader ML community. We will update the introduction and add two new paragraphs "Impact on adoption" in Sections 2 and 4

---

### Official Review · Reviewer_Vkum · 2025-03-23

**Significance:** 2
**Argument Clarity:** 4
**Rating:** 3
**Confidence:** 4

**Questions:**

Questions:
- Within the ICML community, who do the authors consider to be the primary audience for this position paper?
- Could you please clarify precisely *by whom* the broader adoption of causal ML methods is desired, and how the paper substantiates why the proposed principles will lead to broader adoption?
- The paper argues that one should focus on synthetic benchmarks because e.g. semi-synthetic benchmarks have factors fundamentally outside the control of the experimentalist. However, the design space identified by the authors is massive. Thus, a counter-argument could be that semi-synthetic setups “make some of the choices” for the designer based on real data, allowing the user to focus on fewer aspects. The consequences of those ‘choices by nature’ can still be analysed as done in prior works and this paper. Given this and the context of needing to incorporate sufficient realism, why do the authors take the position of focusing on only synthetic experiments?

**Discussion Potential:**

3

**Paper Summary:**

The paper addresses a central conundrum in the evaluation of causal ML methods: namely that, unlike other areas of ML, real-world data with ground truth causal effects are extremely expensive to obtain and curate, yet synthetic data is often dismissed as being overly simplistic or not useful. To this end the paper takes the position that, in order to encourage broader adoption and a more rigorous science of causal ML, synthetic experiments and data are crucial and should be the main means of evaluation of causal ML methods, as opposed to semi-synthetic or real datasets. To substantiate this position, the authors argue and show through examples and empirical studies that semi-synthetic and real datasets inevitably introduce biases (either through human choices or the idiosyncrasies of the dataset) which favor certain causal ML methods over others. In regards to synthetic evaluation, the authors introduce a set of principles as a call to action to the causal ML community: to (i) state all design choices upfront; (ii) move beyond accuracy to more holistic metrics; and (iii) design standardized benchmarks to avoid inconsistent evaluation practices. The paper concludes with an analysis of the potential challenges and alternative perspectives to the stated position.

**Position:**

Yes

**Position In Title:**

Yes

**Related Work:**

4

**Strengths And Weaknesses:**

The paper presents a nuanced discussion of current challenges in evaluation for causal ML methods, and a strong argument for developing more rigorous synthetic benchmarks and how this should be done. As such the paper has high discussion potential and impact for the causal ML community. On the other hand, there is much less substantiation for the claim that this will lead to broader adoption by the ML community. Overall, I favor acceptance as the paper presents well-substantiated and thought-provoking perspectives that will be valuable for the causal ML community.

Strengths:
- **Substantive argument for synthetic benchmarks in causal ML**: The paper provides a well-substantiated argument for focusing on developing more rigorous synthetic benchmarks. The paper references and summarizes existing literature showing how rankings of causal ML methods can change depending on the precise causal mechanism choices in semi-synthetic datasets and experimentally illustrates the same issues with learned mechanisms (RealCause) as well as the need for analysis success and failure modes with previous synthetic evaluations (CausalNF).
- **Actionable principles for causal ML**: Aside from identifying problems in current practice, the paper outlines a number of principles for experimental design in future work on causal ML. Some considerations, such as analyzing the induced distribution of problems constructed by a chosen distribution, and how to perform exact sampling from a well-defined distribution over causal models, queries and datasets are very novel.
- **Consideration of challenges**: The paper does an excellent job in considering the potential challenges associated with the more rigorous synthetic experiments. In particular, the acknowledgement of the additional computational burden of evaluation for the causal ML community, which includes many resource-limited researchers.

Weaknesses:
- **Insufficient specificity of and support for the stated position**: The stated position is that rigorous synthetic experiments are required for broader adoption of causal ML methods. While the authors have made a strong argument for the scientific value of rigorous synthetic experiments, they have not made a sufficiently strong argument for why this would be either necessary or sufficient for broader adoption; and indeed, broader adoption by whom? If the broader ML research community, then a significant unaddressed point is why machine learning researchers (e.g. LLM research) have a need for causal methods, or how causal ML methods will enhance their research and thus convince adoption. If practitioners in applied domains, then there needs to be more specificity or examples of domains where more rigorous synthetic experiments and benchmarks. In my view the paper reads more like identifying best practices for the causal ML community (which I do believe is valuable in itself, if a little narrow), rather than an argument for how to encourage broader adoption of causal ML methods.
- **Addressing the need for realism**: Further, despite the paper acknowledging throughout the introduction and Section 2 that synthetic benchmarks failing to replicate the complexity of real world datasets is a problem, I did not find sufficient arguments later on how this is addressed by the proposed more rigorous synthetic experiments. In particular, *how* should synthetic experiment designers construct generation procedures through your framework that reflect the complexities of reality? This is a crucial point to address as general machine learning is dominated by semi-synthetic benchmarks on downstream applications (e.g. commonsense/mathematical reasoning) with high complexity.

**Support:**

3

---

> ### Author Rebuttal · Authors · 2025-03-31
>
> We thank the reviewer for the constructive review. We are delighted they found the paper impactful for the Causal ML community. Below, we address the raised concerns.
>
> **Q1: "Who is the primary audience?"**: We aim to enrich the debate on rigorously evaluating Causal ML methods to improve current practices. Thus, the paper's primary audience is the Causal ML community and the ML experimental design community. Indeed, we do believe that the best evaluation frameworks will result from the collaboration of both communities, as both causal inference and evaluation expertise need to be combined. We will add this clarification in the introduction.
>
> **Q2: "Why the proposed principles will lead to broader adoption and by whom?"**: Although this is a fascinating topic, this paper does not answer the general question "How to achieve broader adoption of Causal ML?". Indeed, such a question implies a much broader societal consideration that requires humanities and social sciences skills we do not have. Thus, in this work, we do not claim that rigorous synthetic evaluation of Causal ML methods is sufficient for their adoption. Instead, we argue that such an evaluation is necessary for the broader adoption of Causal ML by ML researchers and practitioners.
>
> Causal reasoning enables one to attain a deeper understanding of the world beyond associations. As a result, practitioners and researchers need causal methods to answer causal questions that go beyond simple prediction. Moreover, for researchers or practitioners who are not explicitly seeking to answer causal questions, Causal ML methods are also valuable for improving predictive models. Indeed, in recent years, causal reasoning has brought new perspectives to several ML tasks, leading to the creation of new methods in data augmentation (Teshima & Sugiyama, 2021;  Little & Badawy, 2019) or ML fairness (Makhlouf et al., 2024), to cite a few.
> In addition, the recent paper by Binkyte et al. 2025 reinforces our position by arguing that integrating causal methods into machine learning methods is essential to navigating the trade-offs among key principles of trustworthy ML.
>
> Although the literature has highlighted the theoretical need for researchers and practitioners to use Causal ML methods, one of the major obstacles to its adoption lies in the questioning of its usefulness (Loftus, 2024). Indeed, since Causal ML methods are based on strong and often unfalsifiable assumptions, their practical value is often in doubt. On the one hand, practitioners claim that such hypotheses are unrealistic considering real-world applications. On the other hand, ML researchers, who are used to working with assumption-free methods, criticize the drastic reduction of the application scope to those strictly validating these assumptions. Thus, in his work, Loftus recommends a philosophical shift towards focusing on utility rather than the veracity of models to enable the causal revolution to take place and thus extend the adoption of Causal ML.
>
> In line with Loftus' position, we propose to give it a concrete and actionable dimension based on the definition of the utility of a good, i.e., “its ability to satisfy a particular need” (Cambridge Dictionary). We can adapt this definition to the utility of a Causal ML method as follows: ability (measured using a specific metric) to answer a causal question from a data set under a particular set of assumptions. Consequently, assessing the general usefulness of a Causal ML method means rigorously evaluating it (Principle 1) according to different indicators (Principle 3) to answer a causal question under different sets of assumptions (Principle 2).
> In short, by identifying evaluation best practices, we provide the Causal ML community with the keys to demonstrating the usefulness of causal methods to the wider communities of ML researchers and practitioners, thereby fostering their adoption. We recognize that this line of reasoning needs to be more precisely presented. We, therefore, plan to add “Impact on adoption” paragraphs at the end of sections 2 and 3 to clarify the link between our position on the synthetic rigorous evaluation of Causal ML methods and their adoption.
>
> **Q3: "Why only synthetic experiments?"**: We argue that synthetic data is necessary. However, we do not wish to suggest that they are the only way of evaluating a Causal ML method. Indeed, semi-synthetic data have many advantages, as you pointed out. On the other hand, they also result from Causal ML generation methods that must be rigorously evaluated before use. Our concern relates to temporality rather than significance: we argue that semi-synthetic data generation methods must first be rigorously evaluated by mapping their performance and identifying their potential limitations with synthetic data before they can be used more widely and benefit from their advantages while being aware of the biases inherent in these methods. We will clarify this point in Principle 1.

---

### Decision · Program_Chairs · 2025-04-30

**Decision:**

Accept (poster)

**Comment:**

There is a consensus among the reviewers that this position paper raises an important point about the role of synthetic data in the evaluations of the Causal ML methods, though there are of course a few obvious limitations (e.g., limited choice of method and benchmark, simulation-reality gap, etc). The field of Causal ML has always been limited by the cost of real-world evaluations. Hence, I believe this paper will likely generate fruitful discussions among the causal ML researchers.